# Reproducible flaws unveil electrostatic aspects of semiconductor electrochemistry

Yan B. Vogel[1], Long Zhang[1,2], Nadim Darwish[1], Vinicius R. Gonçales[3], Anton Le Brun[4], J. Justin Gooding [3], Angela Molina[5], Gordon G. Wallace[2], Michelle L. Coote [6], Joaquin Gonzalez[5] & Simone Ciampi[1]

Predicting or manipulating charge-transfer at semiconductor interfaces, from molecular electronics to energy conversion, relies on knowledge generated from a kinetic analysis of the electrode process, as provided by cyclic voltammetry. Scientists and engineers encountering non-ideal shapes and positions in voltammograms are inclined to reject these as flaws. Here we show that non-idealities of redox probes confined at silicon electrodes, namely full width at half maximum <90.6 mV and anti-thermodynamic inverted peak positions, can be reproduced and are not flawed data. These are the manifestation of electrostatic interactions between dynamic molecular charges and the semiconductor's space-charge barrier. We highlight the interplay between dynamic charges and semiconductor by developing a model to decouple effects on barrier from changes to activities of surface-bound molecules. These findings have immediate general implications for a correct kinetic analysis of charge-transfer at semiconductors as well as aiding the study of electrostatics on chemical reactivity.

[1] Department of Chemistry, Curtin Institute of Functional Molecules and Interfaces, Curtin University, Bentley, WA 6102, Australia. [2] ARC Centre of Excellence for Electromaterials Science, Intelligent Polymer Research Institute, University of Wollongong, Wollongong, NSW 2500, Australia. [3] School of Chemistry, Australian Centre for NanoMedicine and ARC Centre of Excellence for Convergent Bio-Nano Science and Technology, The University of New South Wales, Sydney, NSW 2052, Australia. [4] Australian Centre for Neutron Scattering, Australian Nuclear Science and Technology Organization (ANSTO), Lucas Heights, NSW 2234, Australia. [5] Departamento de Quimica Fisica, Universidad de Murcia, 30003 Murcia, Spain. [6] ARC Centre of Excellence for Electromaterials Science, Research School of Chemistry, Australian National University, Canberra, ACT 2601, Australia. Correspondence and requests for materials should be addressed to J.G. (email: josquin@um.es) or to S.C. (email: simone.ciampi@curtin.edu.au)

n 1876 Ferdinand Braun presented to a Natural Society meeting the first deviations from Ohm's law he had observed in crystals of galena, a natural form of lead sulfide. In the following century his discovery revolutionized our civilization. From galena to silicon, materials that can turn from conductors to insulators are at the basis of all our digitized technology[1]. Understanding the full spectrum of factors at play, when charges are transferred across a semiconductor interface is crucial; it underpins the design of devices whose function span from converting light into electricity, to sensing their environment[2], culminating in the very recent report of single-molecule rectifiers on Si(111)[3]. Chemically modified electrodes are therefore a very important laboratory model system[4], however, a search of the literature will indicate that in contrast to metallic electrodes the kinetic parameters for electron transfer at semiconductors are difficult to reproduce from laboratory to laboratory[5, 6]. Widely accepted approaches that are used to gain insights on electrode kinetics have clearly failed to reproduce the complex energetic landscape that determines the redox behavior observed. By highlighting the participation of dynamic electrostatic factors on charge-transfer, we are implicitly demonstrating that the electrostatic landscape of a silicon/molecular layer/electrolyte interface should either be accounted for or eliminated when the focus is on extracting kinetic data at semiconductor or photoconductor electrodes. This knowledge also opens up a semiconductor-based platform to aid the study of electrostatics on chemical reactivity[7–11], and molecular electronics[3, 12, 13].

Here, we study how electrostatic interactions manifest on surface-confined redox molecules and how to gauge these interactions by a general form of electrochemical spectroscopy[14], i.e., cyclic voltammetry. The electrostatic landscape of silicon is particularly rich and the poor Debye screening of the material results in a fraction of the applied bias appearing inside the semiconductor phase itself. This opens up the possibility of interplay between this charged penetration zone within the semiconductor (hereafter referred to as the space-charge region) and a charged layer of surface dipoles[15]. Opposite to the more common situation of a metal-semiconductor contact[16], in this report the excess charge is not localized at step edges or defect sites, but in a redox monolayer outside the semiconductor phase. We have built a phenomenological model to account for electrostatic effects on surface molecules in the presence of faradaic currents at semiconductor/electrolyte systems. This model merges a modified Frumkin isotherm that accounts for both the electrostatic interactions sensed by a molecular layer confined on the electrode surface[17, 18] as well as for the current–potential relationship of the underlying semiconductor photodiode[19] under realistic finite kinetics (Butler–Volmer). Hence, the value of this study is to justify the experimental evidences for electroactive monolayers at Si(111) and to present a general framework to conceptualize charging and current dynamics of semiconductor electrodes. We remark however that the model cannot be directly connected with field values; precise field numbers (V nm$^{-1}$) experienced by surface molecules cannot yet be generated until additional assumptions are validated independently[20].

## Results

### Reproducible electrochemical flaws of a surface model system.
Native silicon always carries a dipole layer of charges on its silica surface and these charges are reflected by a space-charge potential barrier. This barrier, which is electrostatic in nature, is however a very sensitive function of surface conditions, and in actual practice it has no unique reproducible value for a given system[21]. Hence, the first step we took was to limit the possibility of adventitious surface charges by chemically passivating an oxide-

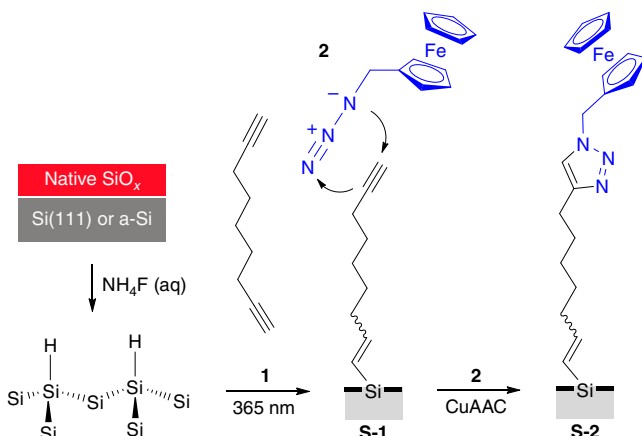

**Fig. 1** The surface model system. Light-assisted hydrosilylation of 1,8-nonadiyne **1** is used to chemically passivate a Si–H electrode and generate an acetylene-terminated monolayer (**S-1**). Azidomethylferrocene **2** is grafted on the electrode via CuAAC click reactions to yield a redox-active film (**S-2**)

free silicon electrode by the hydrosilylation of 1,8-nonadiyne (**1**, Fig. 1)[22–25]. The acetylene-terminated surface (**S-1**) was then modified by the covalent attachment of a reversible redox species, azidomethylferrocene (**2**)[26], with the goal of (i) having control on the surface density of positively charged ferricenium tethers (**S-2** samples) and (ii) analyzing their electrochemical traces to decouple electrostatic effects due to molecule—molecule interactions[27] from molecule—space-charge ones. Figure 2a shows representative cyclic voltammograms obtained for the ferrocene-functionalized Si(111) electrodes (**S-2** on n-type, 7–13 Ω cm, hereafter lowly doped in short hand). The voltammetry was performed under light as the semiconductor electrode is operating in depletion and thereby it requires illumination to carry a current[28] (see Supplementary Fig. 1). This is the conceptual equivalent of the reverse bias in a solid junction, where the current is mainly carried by holes flowing under the barrier (Fig. 2b). The ferrocene-confined probes exhibit broad voltammetry waves (black symbols in Fig. 2a), with the observed full width at half maximum (fwhm hereafter) being on average $142 \pm 6$ mV.

The ideal fwhm from the Langmuir isotherm of a nernstian process is 90.6 mV (model 1, Supplementary Note 1), and theoretical models exist to relate the non-ideal behavior of a voltammetric peak to the balance between attractive and repulsive interactions experienced by a strongly-adsorbed molecule[27]. The increase over the ideal fwhm's is often observed in the literature[29] and can be attributed to a predominance of repulsive interactions between the electroactive species. The apparent kinetics is fast ($k_{et} = 200$ s$^{-1}$, see Supplementary Fig. 2 and Supplementary Note 1 (finite kinetics limit) for details on the calculations of charge-transfer rates) and a quantitative model that takes into account these interactions can be implemented by assuming a nernstian behavior (i.e., very fast electrode kinetics) and Frumkin isotherm, as reported by Laviron in his seminal works (model 2, Supplementary Note 2)[17, 18]. The simulated curves are shown in Fig. 2a as a solid black line. Only the Frumkin parameter $G$ was adjusted in the model; changes to $G$ account for an imbalance in the electrostatic push/pull, forcing the voltammetric wave to broaden or narrow when the activities of the reduced and oxidized species do not follow their surface concentrations. A predominance of repulsive forces in **S-2** samples leads to negative $G$ values of $-0.93 \pm 0.10$ in average. This is expected due to the high density of ferrocenes ($\Gamma = 2.89 \pm 0.24 \times 10^{-10}$ mol cm$^{-2}$, ca. 65% of a full monolayer) and in fact similar observations

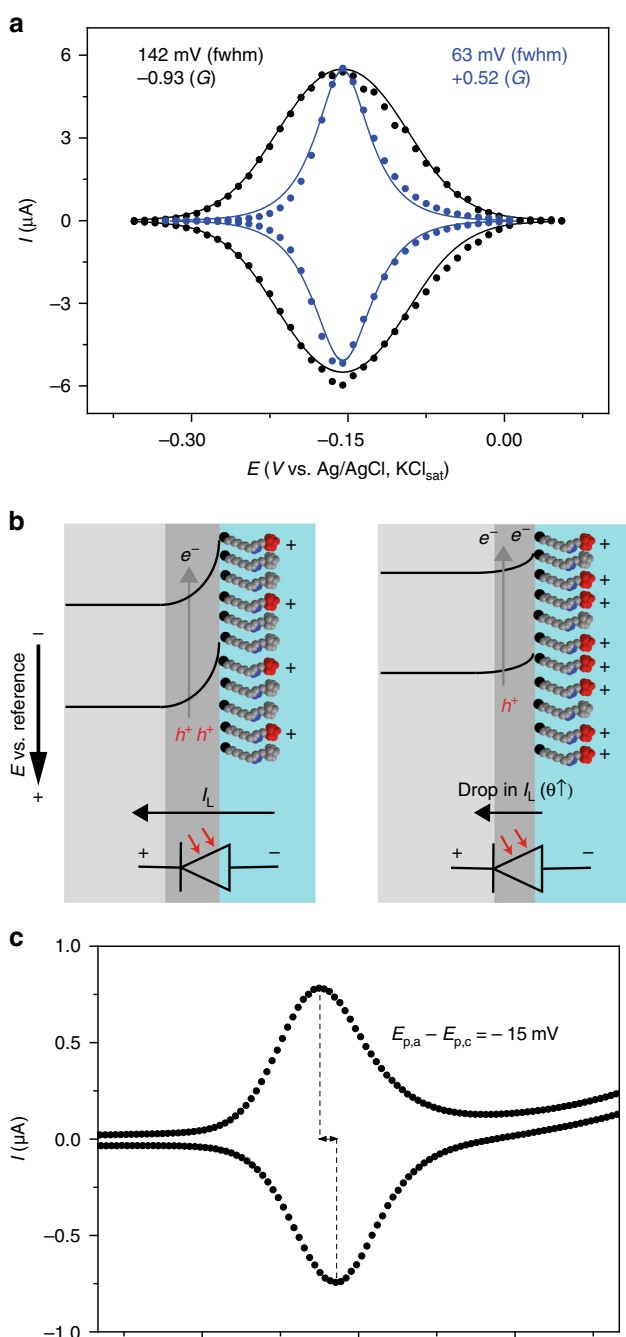

**Fig. 2** Reproducible electrochemical flaws. **a** Representative background-subtracted voltammograms (100 mV s⁻¹, 1.0 M HClO₄) for **S-2** samples on n-type Si(111). Simulated traces (solid black line) for as-prepared samples (symbols) indicate repulsive forces dominate the electrostatic balance of the monolayer system. Applying a potential step of 0.3 V for 140 s shift the balance in favor of attractive interactions (blue line and symbols). Averaged values from cathodic and anodic peaks of experimental fwhm's and refined Frumkin $G$ values appear as labels in figure. **b** Distortion of the semiconductor side of the barrier for a photoanode due to the presence of an electrochemically induced dipole layer of surface charges. The depiction of the photodiode used for the simulations indicates downward flip of the bands due to the positive ferriceniums units and the changes to the photogenerated anodic current. **c** Inverted voltammograms for **S-2** samples (25 mV s⁻¹, n-type). $E_{p,c}$ is 15 mV higher than $E_{p,a}$. The hydrosilylation reaction time (**S-1**) is 10 min

(i.e., large fwhm's and negative $G$ values) are observed for **S-2** samples of a similar coverage prepared on substrates of different conductivity type and level (Supplementary Fig. 3). It is therefore apparent from the broad shape of the voltammograms in Fig. 2a (black traces) that detecting evidence of an electrostatic effect by the space-charge would first require limiting the extent of in-plane ferrocene—ferrocene repulsions and/or decreasing the dielectric screening between the space-charge and the ferrocene (Supplementary Note 3). By means of shortening the click reaction time of azide **2** on samples of **S-1** to a few seconds, the ferrocene coverage can be lowered to $5.9 \times 10^{-11}$ mol cm⁻² (ca. 13% of a full monolayer), however, this has no significant impact on the attraction/repulsion balance (Supplementary Fig. 4) and fwhm's remain above the ideal 90.6 mV.

Very surprisingly, the experimental fwhm's drops reproducibly below the ideal value when the monolayer coverage is being lost by deliberately introducing an oxidative damage (Fig. 2a, blue symbols). Applying a large anodic bias to samples of **S-2** results in the partial loss of the monolayer, with the density of ferrocenes dropping below $1.7 \times 10^{-10}$ mol cm⁻² (i.e., ca. 35% of a full monolayer) and fwhm's unexpectedly dropping to 75–55 mV (Fig. 2a, blue traces, and Supplementary Fig. 5). The experimental voltammograms in Fig. 2a are fitted to yield a $G$ value of +0.52 (solid blue line, model 2), which is diagnostic of attractive forces dominating on the surface tethers. Several lines of evidence relate this remarkable increase in Frumkin $G$ to a space-charge effect. Firstly, to the best of our knowledge this has never been observed on a metallic electrode. Secondly, we discarded that these interactions could involve ionized silicon dioxide (SiO₂); the effect is pH-independent and narrow waves are observed at both pH values of 0.5 and 6.0 (Supplementary Fig. 5). At these values of pH, and neglecting for simplicity fields effects[30], the chemical equilibrium between ionizable surface groups should lead to either positive (SiOH₂⁺) or negative (SiO⁻) surface groups, respectively[31]. Furthermore, a deliberate oxidative damage to samples of **S-2** prepared on highly doped electrodes led to no noticeable effects on $G$ (Supplementary Fig. 6) and thereby it is unlikely that, besides the space-charge of a lowly doped substrate, other obvious sources of electrostatic interactions, such as anions from the electrolyte[27], are contributing toward such a large increase in the self-interaction parameter. AFM images and XPS narrow scans of the Si 2$p$ region in Fig. 3 show the increase in SiO₂ presence and the evolution of surface topography upon the deliberate anodic damaging of **S-2** samples. Voltammograms appear as insets to the AFM micrographs to illustrate that as soon as the high-energy XPS emissions from silicon oxides (Si⁺–Si⁴⁺, SiO$_x$:Si 2$p$ peak area ratio of 0.05) become measurable in Fig. 3e, the voltametric traces narrow below 90.6 mV. The parallel between the spectroscopic appearance of oxides and topographical changes (Fig. 3a–c) is also noteworthy. Figure 3a shows an AFM image taken for an as-prepared **S-2** sample. The surface roughness is extremely low which is an indication of the high quality of the surface samples (ca. 0.2 nm peak-to-valley roughness measured by AFM on individual terraces, details in Fig. 3, or a ca. 0.3 nm overall roughness determined independently by X-ray reflectometry, Supplementary Note 4)[32–34]. The width of the individual surface terraces amounts to ca. 70–100 nm, with the height of the steps being ca. 0.5 nm, which is only slightly higher than a single atomic step between the adjacent lattice planes (0.3 nm). The ⟨111⟩ terraces are also remarkably smooth, with an $R_q$ value of $0.08 \pm 0.02$ nm ($R_{tm} = 0.15$ nm), however a number of small protrusions can be observed (ca. 200 protrusions μm⁻²), which are on average 0.4 nm in height and 5 nm in radius. An extensive anodic treatment (Fig. 3c, f, SiO$_x$:Si 2$p$ of ca. 0.1) leads to a rougher surface ($R_q$ of $0.11 \pm 0.01$ nm ($R_{tm} = 0.23$ nm), Supplementary Figs. 7 and 8) and the number

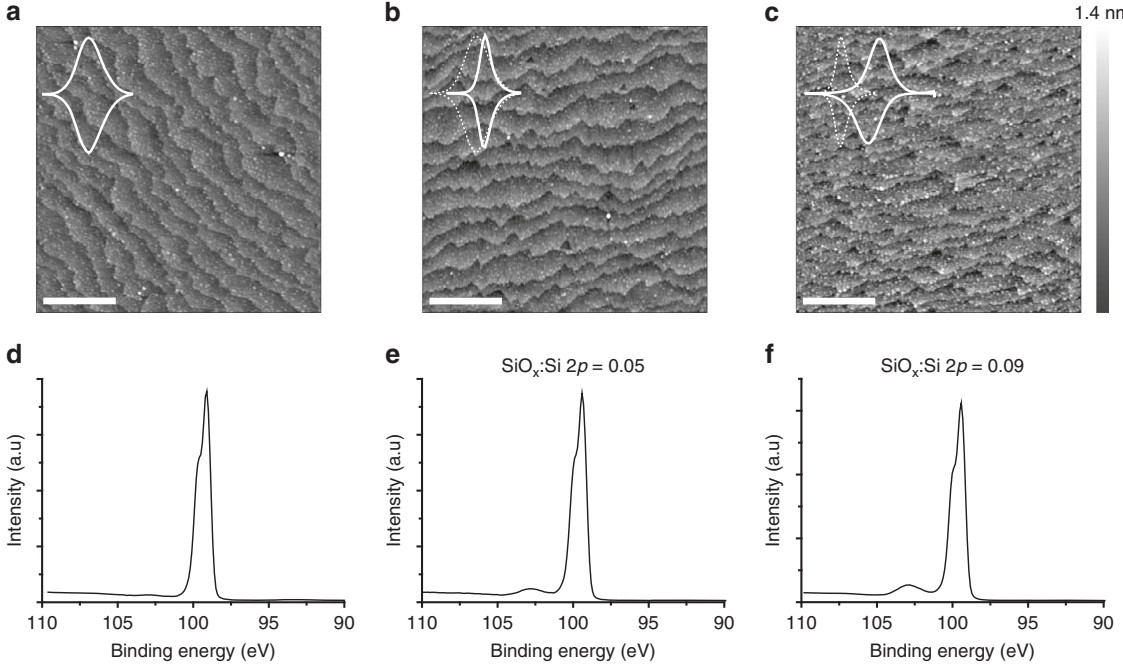

**Fig. 3** Anodic monolayer stripping. Tapping mode AFM images (2 × 2 μm) and XPS Si 2p narrow scans for **S-2** samples on Si(111). Cyclic voltammograms are shown as insets to the AFM images. Data for as-prepared samples are in **a** and **d**, and data after the short potential step (0.3 V, 1 min) that lead reproducibly to narrow voltammetric waves (ca. 60 mV fwhm) are in **b** and **e**. Owing to the kinetic factors, the narrow waves are lost upon more extensive oxidation of the electrodes (**c**, **f**). High-energy emissions from silicon oxides (Si$^+$–Si$^{4+}$) are evident in **e** and **f**, and the evolution of the SiO$_x$:Si 2p peak area ratios appear as labels in the XPS panels. **a** Small protrusions are observed (222 protrusions μm$^{-2}$, which are on average 0.4 nm in height and 5 nm in radius) over the staircase structure of the ⟨111⟩ surface ($R_q$ of 0.08 ± 0.02 nm ($R_{tm}$ = 0.15 nm) on the terraces and $R_q$ = 0.20 ± 0.15 nm ($R_{tm}$ = 0.63 nm) over the whole area). The number of protrusions increases upon the anodic treatment (327 protrusions μm$^{-2}$ in **b** and 945 protrusions μm$^{-2}$ in **c**) and tracks XPS SiO$_x$ emissions. For clarity and comparison purposes all the AFM images are normalized to 1.4 nm. The scale bar is 500 nm

and size of protrusions increases (ca. 900 protrusions μm$^{-2}$, 0.5 nm in height and 8 nm of radius). However, if the anodic treatment is milder (SiO$_x$:Si 2p of ca. 0.05), just to suffice in narrowing peaks below 90.6 mV (inset to Fig. 3b), there are no measurable changes to surface $R_q$ and $R_{tm}$, (Fig. 3b; Supplementary Fig. 7) but only an increase to the number of nanometer-sizes protrusions (ca. 300 protrusions μm$^{-2}$, Fig. 3b; Supplementary Fig. 7). It has been suggested by Allongue and co-workers[32] that the number of these rounded features is possibly related to silica islands that can be measured by AFM even when the XPS SiO$_x$ emission is below the detection limit (Fig. 3a).

**Inversion of peak potentials**. As for the current-potential data shown in Fig. 2a, the near-surface barrier of the electrode can be transiently lifted by illumination in the visible[28, 35]. Even a strong illumination will only minimize without nulling the semiconductor barrier, hence illumination—which is required to get a current output—may not totally compromise the semiconductor's ability to sense or exert electrostatic interactions on nearby molecules. Figure 2b shows the conventional energy-band representation that describes a plausible overall effect of an externally applied electric field and a dipole layer of surface charges on the semiconductor barrier. The convention is to draw energy-levels diagrams such that the energy of the system is lowered as electrons fall down toward the source of positive potential (i.e., drawn at the bottom of the diagram). The experimental observation of voltammetric fwhm's below 90.6 mV and a refined positive $G$ indicates the presence of attractive forces on the positive ferricenium tethers, which can be conceptualized as electrons redistributing over the semiconductor side of the barrier. We should therefore analyze this hypothesis against its

most obvious consequence. According to the convention on energy-level diagrams, the higher the semiconductor side is raised by the static positive charges of the ferricenium ions, that is to say moving from the left to the right panel of Fig. 2b, the more (photo)electrons there would be at an energy above the top of the barrier (i.e., increasing reduction rates). At the same time, this distortion of the bands, induced by the presence of the dipole layer of surface charges, should lift the opposition current carried by holes (i.e., decreasing oxidation rates). The direct consequence would be a very intriguing inversion of peak positions (Supplementary Note 5), with the potential for the cathodic peak shifting anodic of the anodic maxima ($E_{p,c} > E_{p,a}$)[27]. The narrowing of the peaks below 90.6 mV and the inversion of the peak potentials are therefore the anticipated manifestation of the same electrostatic process. Interestingly, only narrow but not inverted peak potentials are observed consistently. In our lowly doped n-type system, **S-2** samples systematically showed inverted peak potentials (5–55 mV, Fig. 2c; Supplementary Fig. 9) only if the hydrosilylation reaction time to prepare **S-1** samples was lowered from 2 h to either 10 or 2 min (Supplementary Figs. 9–11). Notably, short times for the passivation step also lead to the inversion and narrow waves, and with no requirements of an oxidative pre-treatment, but the phenomena is however short-lived and does not allow for systematic changes to the voltage sweep rate and hence rigorous data modeling (Supplementary Fig. 9c). We speculate that a low coverage of diyne **1** molecules after a short 2 or 10 min reaction suffices to reduce the dielectric screening of the carbonaceous film (Supplementary Fig. 12)[36], thereby allowing for inversion, but on the other hand the poor passivation leads to the growth of SiO$_2$ which masks the inversion by reducing the electron transfer kinetics (*vide infra*). It is in fact very plausible that the manifestation of peak potential inversion is

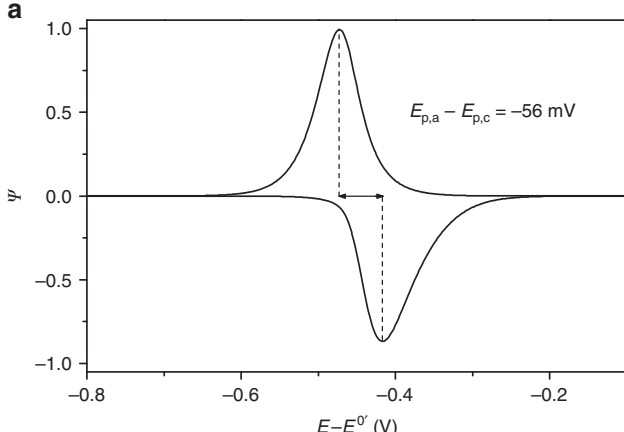

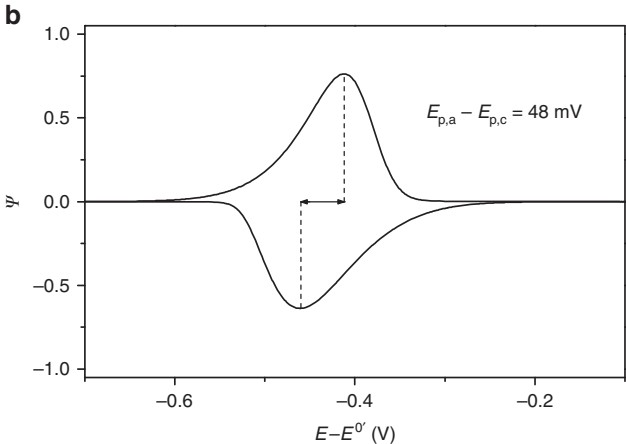

**Fig. 4** Kinetic factors mask the inversion of peak potentials and the true peak dispersion. Simulated voltammograms (25 mV s$^{-1}$) showing the masking of the system's electrostatic (i.e., diode and Frumkin effects) by slow kinetics. Curves are calculated using model 3 and by setting $k_{et}$ to either 100 s$^{-1}$ (**a**) or 1 s$^{-1}$ (**b**). Frumkin G was + 0.5 and $\theta$ is adjusted from 0.04 in the anodic segment to 1.95 in the cathodic one. This increase in $\theta$ reflects the ca. 50-fold drop in the (photo)hole opposition current ($I_L$) generally required to fit experimental data. The y-axis current is expressed as $\psi = \left(I/I_{p,rev}\right) \times \left(RT/(Q_F Fv)\right)$, with $I_{p,rev}$ being the peak current obtained for a fast charge-transfer process (see Supplementary Note 6 and Supplementary Fig. 14 for details)

masked to some degree by kinetic factors, which forces a progressive peak-to-peak separation ($E_{p,a} > E_{p,c}$). To quantitatively reinforce this argument, we have developed a model of a system that assumes finite kinetics (Butler–Volmer), accounts for Frumkin's interactions of the surface tether and includes dynamic parameters describing diode effects for the potential and currents across the space-charge (model 3, Supplementary Note 6). Parameters for the diode element can be determined in the reversible region and can yield quantitative information on the magnitude of reverse saturation current and photocurrent. Changes to these two values, or more precisely how the value of $\theta$ decreases in response to an increase in the surface concentration of positively charged ferricenium units is crucial to the understanding of the inversion effect, where $\theta$ is given by Equation (1):

$$\theta = I_{peak}/(I_L + I_0) \qquad (1)$$

where $I_{peak}$ is the peak current height in the voltammograms, $I_L$ is the diode photocurrent and $I_0$ is the diode reverse saturation current, see Supplementary Fig. 13.

Simulations to illustrate how changes in $\theta$ are reflected in the peak splitting when the kinetics of electron transfer is either fast or slow are shown in Fig. 4a and b, respectively. For fast kinetics ($k_{et}$ of 100 s$^{-1}$) that is coupled to a Frumkin G of +0.5 our model predicts the inversion to be clearly evident in voltammograms if $\theta$ increases from 0.04 in the anodic segment to 1.95 in the cathodic one. This change in $\theta$ is the result of bands flattening to accommodate for the electrostatics in the system (Fig. 2b), it indicates a ca. 50-fold drop in the (photo)hole opposition current ($I_L$, model 3), and it manifests as inverted peak potentials (Fig. 4a). For the same G value and the same change in $\theta$ but under a slower kinetics, for instance with $k_{et}$ dropping to 1.0 s$^{-1}$, the peak order would revert to an apparent normal situation (i.e., $E_{p,a} > E_{p,c}$, Fig. 4b). The apparent $k_{et}$ for the narrow blue trace in Fig. 2a is ca. 80 s$^{-1}$ (see also Supplementary Fig. 2), but upon more extensive oxidative damage of the sample (Fig. 3c, f; Supplementary Fig. 2), this value declines further towards the irreversible region ($k_{et} = 17$ s$^{-1}$). At this point both experimental data and simulations (Supplementary Figs. 2, 14 and 15) indicate that the electrostatic pull between the space-charge and surface-bound molecules, which leads to narrow waves and inversion, is completely masked by kinetics. As a consequence of the positive charges on the surface tethers, bands in the semiconductor are distorted downward (Fig. 2b). Using the model 3 (Supplementary Note 6, finite kinetics limit), we have accounted for the inverted peak potentials in n-type **S-2** samples and rationalized this as an electrostatics-induced decrease in the anodic photocurrent in the reverse scan (Fig. 2c; Supplementary Figs. 9 and 13).

**Electrostatic effects on diode and kinetic analysis**. If our reasoning is correct, and electrostatics is indeed a major factor at play, it is thereby easy to see that a similar inversion effect ($E_{p,c} > E_{p,a}$) would also hold for p-silicon photocathodes (Fig. 5a, $I_L$ here increases in the reverse scan, vide infra). The technical problem here is that the chemical system (i.e., **S-2**) we are using to sense the electrostatics of the solid/liquid interface has a relatively anodic formal potential, and it would therefore fall in the accumulation regime of a p-type electrode. To get around this problem we prepared control **S-2** samples on thin films (ca. 4 μm) of amorphous silicon (a-Si hereafter) that are grown on highly doped p-type substrates. The a-Si electrodes used in this study are near-intrinsic photoconductors, but the nature of the p-type back-contact forces the interface to act, to some degree, as a photocathode (Supplementary Fig. 16). Figure 5b and Supplementary Figs. 17 and 18 show cyclic voltammograms for **S-2** samples on a-Si. It is apparent that on the a-Si surface the redox tether experiences analogous attractive electrostatic forces as observed for the n-type Si(111) samples. The interactions between charged ferricenium cations and the semiconductor side of the barrier (Fig. 5a) consistently leads to narrow waves and inverted peak potentials, but unlike the crystalline Si(111) system, the effect is already apparent in as-prepared samples (Fig. 5b, black trace), it does not require a short anodic pulse (Fig. 2a) and it is long-lived (Supplementary Fig. 18). A rigorous explanation on these aspects—as why the manifestation of electrostatic is more pronounced and more robust in rough amorphous samples (Supplementary Fig. 19)—is beyond the scope of this work, but at this stage it is possible to speculate on a relationship between disorder in the film and kinetics factors[37]. The electrochemical non-idealities (i.e., narrow and inverted waves) in a-Si samples are sustained over a prolonged analysis; hence, it is possible to apply model 3 (Supplementary Note 6) to its full extent. At voltage sweep rates of about 100 mV s$^{-1}$ we can observe a drift from the inverted region (Fig. 5b, black traces, and Fig. 6) to a normal region, where the shift caused by redox kinetics masks the

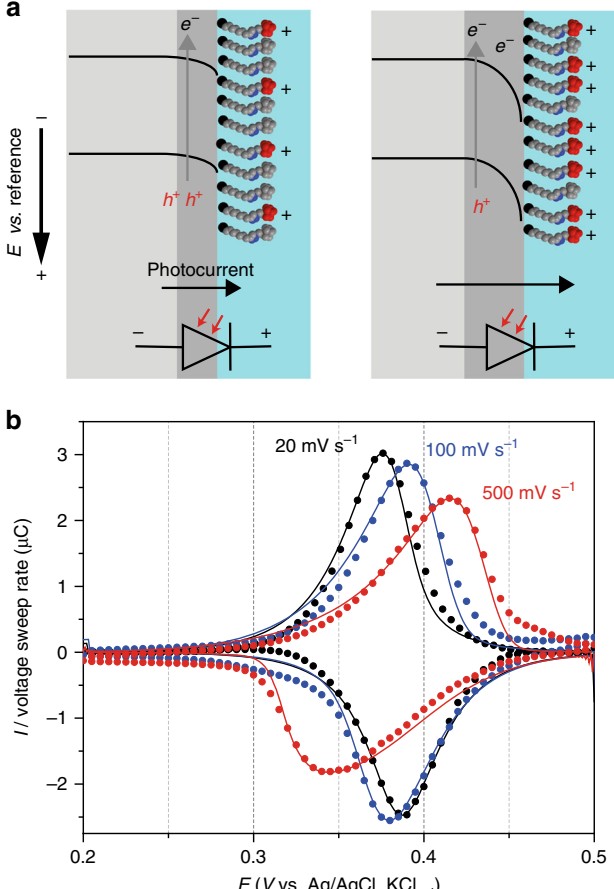

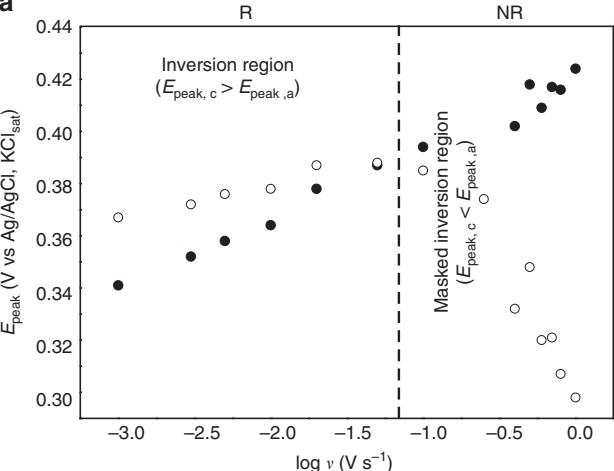

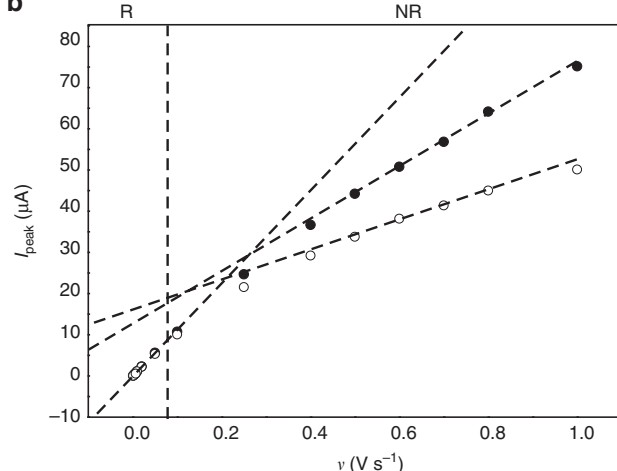

**Fig. 5** Validation of the model and its implications for the correct kinetic analysis of charge-transfer reactions at semiconductors. **a** Energy-level diagram depicting the distortion of the semiconductor side of the barrier for a-Si photocathodes due to electrochemically induced changes to surface charges. **b** Experimental (symbols) and simulated voltammograms (solid lines, model 3, Supplementary Note 6) for **S-2** samples over a wide range of voltage sweep rates (20–500 mV s$^{-1}$). Simulation parameters are listed in Table 1

**Fig. 6** Kinetic regions for the voltammetry as a function of the scan rate. Evolution of the experimental peak potentials ($E_{peak}$, **a**) and currents ($I_{peak}$, **b**) as a function of voltage sweep rate ($\nu$) in the cyclic voltammetry of **S-2** samples on a-Si. Black and white symbols correspond to the anodic and cathodic data, respectively. Reversible (fast kinetics, R), and non-reversible (finite kinetics, NR) zones are qualitatively indicated in figure. Lines are a guide to the eye only

electrostatic effect (Fig. 5b, blue and red traces). In Fig. 6, we have tentatively marked two regions relative to the scan rate (R as short hand for reversible and NR for non-reversible, Supplementary Note 7) and it is clear that the reverse peak suffers the strongest influences of both diode and kinetics. In the a-Si system the photodiode element is expected to point towards the electrolyte and the photocurrent is cathodic (Fig. 5a; Supplementary Fig. 20).

Model 3 combines the effects of diode (through photocurrent and reverse saturation current) plus finite kinetics for the charge-transfer reaction at the monolayer side (through the corresponding rate constant and charge-transfer coefficient in the B–V formalism), and presence of interactions (through the Frumkin interaction parameters). The model appears therefore to be rather complex, but nonetheless the manifestation of the electrostatic effect remains simple to diagnose in terms of narrow and inverted waves. The parameters used to refine theoretical curves (Fig. 5b, solid lines) against experimental data (symbols) are given in Table 1. From the data in table it can be concluded that for small potential sweep rates (e.g., 20 mV s$^{-1}$), there is no significant perturbation caused by the redox kinetics on the electrostatic of the system. As a consequence of the electrostatics experienced by the space-charge of the electrode the photocurrent value for the reverse scan is one order of magnitude greater that in forward

**Table 1 Interaction parameters obtained from current-potential curves in Fig. 5b**

| Scan rate (mV s$^{-1}$) | Direct scan | | | | Reverse scan | | | |
|---|---|---|---|---|---|---|---|---|
| | $Q_F$ | $G$ | $s$ | $y$ | $Q_F$ | $G$ | $s$ | $y$ |
| 20 | 6.9 | 1.0 | 0.15 | 0.2 | 6.1 | 1.0 | -0.9 | 0.2 |
| 100 | 7.5 | 1.0 | -0.2 | 0.2 | 6.5 | 0.9 | -0.8 | 0.2 |
| 500 | 6.7 | 1.2 | -0.7 | 0.1 | 6.7 | 0.8 | -0.6 | 0.2 |

Refined parameters are $k_{et} = 80$ s$^{-1}$, $\alpha = 0.5$, $I_0 = 10^{-5}$ µA, $I_L$ (direct scan) = 10 µA, $I_L$ (reverse scan) = 100 µA. $Q_F$ is in µC and defined as the product $FAT_T$. The interaction parameters can be calculated as $a_{or} = (y-G)/2$, $a_r = (s+y)/2$, $a_o = -(s-y)/2$

segment ($I_L$ (reverse scan) = 10 × $I_L$ (direct scan)), hence the inversion of the current peak on the potential. An increase in the voltage sweep rate, $\nu$, leads to a decrease of the effective rate constant ($k_{et}\nu^{-1}$) and therefore to an enhancement of the influence of kinetics on the response, causing a shift of the direct/reverse curves towards more positive/negative potentials, respectively, which in turn masks the potential inversion.

Figure 6a and b (and Supplementary Fig. 21) are plots of the experimental peak potentials and currents values obtained over a broader range of voltage sweep rates. For small values of $\nu$, the peak potential inversion is obvious, and both the anodic and cathodic current maxima depend linearly on $\nu$ (R region corresponding to a fast kinetics, with $\overline{k}_{et} = k_{et}/(F\nu/RT) \geq 20$). An increase in the voltage sweep rate leads to a progressive nearing of the direct and reverse curves arising from a kinetic distortion of the response for $20 < \overline{k}_{et} < 5$ with $E_{p,a} \approx E_{p,c}$ and loss of the linear relationship between peak currents and $\nu$. For higher scan rates, $\overline{k}_{et} < 5$ (NR region), kinetics dominate the response but it is noteworthy that to some extent diode effects are still present. Although peak currents show a linear dependence with $\nu$, the anodic and cathodic slopes are different and the reverse peak is much more affected than the forward one. This aspect is also reflected in the experimental fwhm's, which do not coincide with those predicted by considering kinetics only (Supplementary Fig. 15).

## Discussion

This study highlights the previously overlooked importance of the electrostatic interactions between a molecular layer and excess charges in the space-charge of a semiconductor electrode. We show how these interactions do manifest at a Si(111)/liquid electrolyte interface and under what circumstances they dominate dynamic currents to and from a surface tether. We demonstrate that electrochemical non-idealities cannot be overlooked and discarded as flaws (fwhm's < 90.6 mV and inverted peak potentials, $E_{peak\ cathodic} > E_{peak\ anodic}$). A model is developed to account for these observations based on the relative weight of diode currents, the balance between static attractions and repulsions, and kinetic factors for the electron transfer reaction. It expands on previous work devoted to account for molecular charges on the potential profile at metallic electrodes[38], showing that the inclusion of dynamic changes to the local energy of bound molecules (i.e., changes to activation free energy of the reaction linked to the presence of intermolecular interactions as described by Frumkin isotherm) and dynamic changes to space-charge effects (i.e., changes to diode parameters) are crucial for a suitable description of semiconductor electrodes. It has immediate implications for the study of electrode kinetics at semiconductors and photoconductors since the electrostatics of the space-charge on surface-tethered molecules can either be prevented or enhanced at will by modifying, for instance, the dielectric of the surrounding environment, here exemplified by changes to the surface coverage of the organic monolayer upon electrochemical cleavage. The cautionary note is that the established kinetic models based on the analysis of peak positions in current-potential traces must either be revised to take into account electrostatics, or the experimentalist must take precautions to limit this type of effects.

The models described are also intended to guide the development of the experimental platforms for the study of how charged groups or externally applied electric fields can influence chemical bonding and reactivity, an area that is beginning to attract significant interest in chemical catalysis[7, 10, 11, 39–41]. This electrostatic aspect of chemistry has been long suggested by theoreticians, with models developed and refined by Shaik[42–45] and others[9, 46] starting from 1981, but until recently this has remained mainly a theoretical exercise[7]. Some progress has been made on insulators, where Kanan and co-workers have demonstrated the effect of static electricity on carbene reactions and epoxide rearrangements at insulators/electrolyte interfaces using elegant surface chemistry on insulating $Al_2O_3$ films[47, 48]. The use of an electrical insulator ($Al_2O_3$) blocks the flow of faradaic currents, and, at least in part[49], it removes complications from redox side reactions. There are however two caveats with insulators. Firstly, insulators can indeed gain excess surface charging[30, 50], but in a material that by its very nature does not conduct electricity it is hard to define, control and measure these effects systematically. These tasks can also be further complicated by surface ionization and adventitious adsorption reactions[51]. Secondly, and most importantly, the use of an insulator compromises a priori the exploration of electrostatic effects over chemical processes that involve a mixed sequence of redox and non-redox steps[52], or the possible use of redox switches[53, 54] together with external electric fields[55] to control plasmonic resonances[56]. A key example of the former case, still awaiting an experimental scrutiny, is cytochrome P450. Intriguing theoretical predictions by Shaik suggest an external oriented field could be used to promote both the non-redox gating as well as the two reduction steps in the cycle of P450; overall increasing at will the enzyme's efficiency[52]. The experimental insights reported in this paper will help to decouple the electrostatic from the dynamic electrochemical process; most importantly allowing one to detect the presence of residual static charges while redox currents are allowed to flow.

## Methods

**Chemicals and materials**. Unless stated otherwise all chemicals were of analytical grade and used as received. Hydrogen peroxide (30 wt% in water), sulfuric acid (Puranal$^{TM}$, 95–97%), ammonium fluoride (Puranal$^{TM}$, 40 wt% in water), ammonium sulfite monohydrate (($NH_4$)$_2SO_3$, 92%), and 1,8-nonadiyne (**1**, 98%) used in wafers cleaning, etching, and silicon modification procedures were obtained from Sigma-Aldrich. Redistilled solvents and Milli-Q water (>18 MΩ cm) were used for substrate cleaning, surface modification procedures, and to prepare electrolytic solutions. Azidomethylferrocene (**2**) was prepared from (hydroxymethyl) ferrocene through published methods[26]. Prime grade 111-oriented (⟨111⟩ ± 0.5°) silicon wafers were obtained from Siltronix, S.A.S. (Archamps, France). The wafers were polished on one side only, were 180–220 μm in thickness and were either n-type (phosphorous) or p-type (boron) doped. Two different grades of phosphorous doping were used, with the resistivity specified by the manufacturer being either 7–13 Ω cm (hereafter referred to as lowly doped Si(111)), or 0.003–0.007 Ω cm (referred to as highly doped). Boron-doped samples were 500 μm thick, 0.03–0.07 Ω cm, and were referred to as highly doped p-Si. Amorphous hydrogenated silicon films were ~4 μm in thickness and 4.2 nm in roughness (see Supplementary Fig. 19) and were prepared by decomposition over a hydrogenated p-type silicon substrate (boron-doped, 100-oriented, (⟨100⟩ ± 0.5°), 0.001–0.003 Ω cm resistivity) of silane gas (SiH$_4$) in an AC plasma (13.56 MHz, 300 W). The amorphous samples are referred to as a-Si.

**Electrode preparation**. Silicon samples were mechanically cut into ~1 cm$^2$ pieces, cleaned under a stream of nitrogen gas, rinsed several times with small portions of dichloromethane and water, cleaned in hot Piranha solution for 20 min (100 °C, a 3:1 (v/v) mixture of concentrated sulfuric acid to 30% hydrogen peroxide), rinsed thoroughly with water and immediately etched with an argon-saturated 40% aqueous ammonium fluoride solution for 10–12 min under an argon atmosphere. The etching bath was added with a small amount (ca. 5 mg) of ammonium sulfite. The freshly etched samples were washed sequentially with water and dichloromethane and blown dry in argon before the dropping of a small deoxygenated sample of diyne **1** (ca. 50 μl) on the wafer. The liquid sample was contacted with a quartz slide to limit evaporation, rapidly transferred to an air-tight and light-proof reaction chamber, and kept under positive argon pressure. A collimated LED source ($\lambda$ = 365 nm, nominal power output >190 mW, Thorlabs part M365L2 coupled to a SM1P25-A collimator adapter) was fixed over the sample at a distance of ca. 10 cm. After illumination for a 2 h period (unless stated otherwise, see Supplementary Figs. 9, 10 and 12) the acetylene-functionalized samples (**S-1**, Fig. 1) were removed from the reaction chamber, rinsed several times with dichloromethane and rested in a sealed vial under dichloromethane at 4 °C for 8 h. The acetylene-terminated samples (**S-1**) were then rinsed several times with small amounts of 2-propanol and transferred to a reaction tube containing (i) the ferrocene molecule (**2**, 0.5 mM, 2-propanol/water, 1:1), copper(II) sulfate pentahydrate (20 mol % relative to **2**) and sodium ascorbate (ca. 100 mol % relative to **2**). Unless stated otherwise (see Supplementary Fig. 4), the click copper-catalyzed alkyne–azide cycloaddition reaction (CuAAC[57] in short hand) was carried out in the dark, at room temperature under air and stopped by removing the ferrocene-functionalized samples (**S-2**) from the tube after a reaction time of 30 min. Samples (**S-2**) were rinsed sequentially with copious amounts of 2-propanol, water, 0.5 M aqueous hydrochloric acid, water, 2-propanol, dichloromethane, and blown dry under nitrogen before being analyzed.

**Surface characterization**. *X-ray photoelectron spectroscopy*. X-ray photoelectron spectroscopy measurements were performed on a Kratos Axis Ultra DLD spectrometer using a monochromatic Al-Kα (1486.6 eV) irradiation source operating at 150 W. Spectra of Si 2*p* (90–110 eV), C 1*s* (277–300 eV), N 1*s* (390–410 eV), and Fe 2*p* (690–740 eV) were taken in normal emission at or below $7 \times 10^{-9}$ Torr. Data files were processed using CasaXPS© software and the reported XPS energies are binding energies expressed in eV. After background subtraction (Shirley), spectra were fitted with Voigt functions. To correct for energy shifts caused by adventitious charging all peak energies were corrected with a rigid shift to bring the C 1*s* emission to 285.0 eV. See Supplementary Methods for a detailed analysis of the XPS measurements.

*Atomic force microscopy*. Surface topography was imaged with a Bruker Dimension atomic force microscope. All images were obtained in air at room temperature and using silicon nitride cantilevers (TESPA-Bruker AFM probes, with spring constant of 20 N m$^{-1}$). The scan area was set to between $3 \times 3$ and $2 \times 2$ μm with a resolution of 512 points/line. The surface roughness was determined using Bruker's Nanoscope Software by measuring the average root mean square, $R_q$, deviation in height of at least five areas on each sample. Values of peak-to-valley roughness, $R_{tm}$ are also reported. The 95% confidence limit of the mean ($x$) on the $R_q$ is reported as $x \pm t_{n-1}s/n^{1/2}$[58], where $t_{n-1}$ depends on the number of repeats and was varied between 2.78 and 2.26, $s$ is the standard deviation, and $n$ is the number of measurements ($n$ was between 5 and 10).

*Specular X-ray reflectometry*. Specular X-ray reflectometry was measured under ambient conditions at the solid—air interface using a Panalytical X'Pert Pro X-ray reflectometer. The X-ray beam was focused and collimated using a Göbel mirror and a 0.1 mm wide pre-sample slit. Specular reflectometry (angle of incidence = angle of reflection) was collected at glancing angles with an incident angle range of 0.05° to 5.00° using a step size of 0.01°. The counting time for each step was 7 s. The data were reduced by normalizing the raw data so that the critical edge was unity and was presented as reflectivity ( = reflected intensity/incident intensity) vs. momentum transfer ($Q$) which is equal to $4\pi \sin\theta/\lambda$, where $\theta$ is the angle of incidence and $\lambda$ is the X-ray wavelength (1.54 Å). Structural parameters for the monolayer were refined in MOTOFIT reflectometry analysis software[59].

**Electrochemical characterization**. Electrochemical experiments were performed in a single-compartment, three-electrode PTFE cell with the modified silicon surface (S-2) as the working electrode, a platinum mesh as the counter and a silver/silver chloride in saturated potassium chloride as the reference electrode. All potentials are reported vs. the reference electrode. All aqueous solutions for electrochemical experiments contained 1.0 M of either NaClO$_4$ or HClO$_4$ (with a pH of 6.0 and 0.5, respectively). The surface coverage, $\Gamma$, expressed in mol cm$^{-2}$, was calculated from the faradaic charge taken as the background-subtracted integrated current from the anodic scan of the voltammograms. Unless specified otherwise, all electrochemical experiments were performed in air at room temperature ($22 \pm 2$ °C) under the illumination provided by a collimated 625 nm LED source (nominal power output > 770 mW, Thorlabs part M625L3, coupled to a SM1P25-A collimator adapter). The incident light was pointed on the top-side (i.e., electrolyte side) of the working electrode. Ambient light was used for the highly doped n-type and for p-type samples. A rectilinear cross-sectional Viton gasket defined the geometric area of the working electrode to 0.28 cm$^2$. Ohmic contact to the working electrode was ensured by gently grinding with emery paper a thin layer of gallium-indium eutectic over the back side of the wafer. A copper plate was pressed against the eutectic. Electrochemical measurements were performed using a CHI650D electrochemical workstation (CH Instruments, Austin, TX). The 95% confidence limit of the mean of experimentally determined quantities, such as $\Gamma$ and fwhm's is reported as $t_{n-1}sn^{-(1/2)}$, where $t_{n-1}$ was varied between 4.30 and 2.45, and $n$ was between 3 and 7.

**Simulation methods**. The experimental cyclic voltammograms of S-2 samples showed non-idealities that were explained by comparison of the data against simulated current-potential responses. Three simulation models were used in this work. These are described only briefly below, and in more detail in the Supplementary Notes 1, 2 and 6. The simulations for models 1 and 2 were based on published works[17, 18, 55, 60, 61] and were written and performed in MATLAB®. Model 3 accounts for finite kinetics and for the presence of attractive and repulsive electrostatic forces on space-charge diode and it was programmed in Mathcad® 14.0. Model 1—The Langmuir model for a single charge-transfer (Supplementary Note 1)[61]. This model describes the voltammetric responses for electroactive surface adsorbed molecules on metallic electrodes when either fast or finite kinetics (Supplementary Note 1 and 2, respectively). Model 2—The interaction model for a single charge-transfer (Supplementary Note 2). The model includes parameters describing interactions between the adsorbed molecules, which are reflected in changes to fwhm's values and peak positions (Supplementary Fig. 22). Model 3—The diode model with Frumkin interactions (Supplementary Note 6). We developed a phenomenological model to accommodate for electrostatic forces involving electroactive molecules, space-charge diode effects and fast or finite kinetics (Butler–Volmer treatment, B–V in short hand notation, Supplementary Fig. 14). We note that in this work the schematics on band-diagrams and diode elements (space-charge) are drawn with the arrows showing the flow of electrons (IUPAC

convention). Tilting of the energy-level diagram downward toward the positive end of the crystal/electrolyte system is neglected for clarity.

**Code availability**. Scripts are available from the authors upon request.

**Data availability**. The data supporting the findings of this study are available from the corresponding authors upon request.

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

## Acknowledgements

This work was supported by grants from the Australian Research Council (ARC, DE160100732 (S.C.), DE160101101 (N.D.)). J.G.S. and A.M. greatly appreciate the financial support provided by the Fundación Séneca de la Región de Murcia (Projects 19887/GERM/15 and 18968/JLI/13) and by the Ministerio de Economía y Competitividad (projects CTQ-2015-65243-P and CTQ-2015-71955-REDT Network of excellence "Sensors and Biosensors"). L.Z., M.L.C., and G.G.W. acknowledge funding from the ARC Centre of Excellence Scheme (Project No. CE 140100012). J.J.G. and G.G.W. are under ARC Laureate Fellowships (FL150100060 and FL110100196). We thank Dr. Jean-Pierre Veder from the John de Laeter Centre for the assistance with XPS measurements.

## Author contributions

All authors contributed to conceiving the work and designing and discussing the experiments. Y.B.V. conducted most of the surface and electrochemical experiments with assistance from S.C. and L.Z. N.D. and Y.B.V. carried out the AFM experiments and analyzed the data. J.G. developed the theoretical models and codes and performed the treatment of the electrochemical data with comments and suggestions from A.M. Y.B.V. and S.C. collected and analyzed the XPS data. A.L.B. collected and analyzed the XRR data. S.C. wrote the manuscript with significant contributions from all the other authors.

## Additional information

**Competing interests:** The authors declare no competing financial interests.

