## [Peer Review File · Nature Communications]

Reviewers' comments:

Reviewer #1 (Remarks to the Author):

This paper report a very detailed study on the electrochemical behavior of molecular monolayer on Si/SiO₂ surface. Some findings from this work is interesting and convincing, including the role of static effects and the inversion of peaks in cyclic voltammograms. On the other hand, I have to conclude that it seems the paper may not be of interest to general audience, and I could not find sufficient significance for the observation reported in this paper.

Besides the significance of the paper, I have also some comments and suggestions.

1. Concerning the selection of surface model system, what is the purpose to induce the long alkane chain? Will the length influence the conclusion?
2. The z scale of Figure 3a-c the surface is missing. Concerning the coverage calculation, as suggested by the AFM images, the surface seems quite rough. I was wondering whether the surface roughness calculation from average root mean square is valid for this case. Some control experiment may be required.
3. In figure 6, it seems almost not data point presented in the QR regime, why? Some further discussion will help.
4. I can not quite follow the schematic of Figure 3b and 5a, especially the E vs reference arrow, what changed?

To conclude, I think this paper is interesting but may not provide enough significance for publication in nature communications. I have the feeling that this paper could be more suitable for a more specific journal.

Reviewer #2 (Remarks to the Author):

A report on

"Reproducible flaws unveil electrostatic aspects of semiconductor electrochemistry" Nature Communication

This paper reports some important science that can be a proper guidance and will be a reference for an emerging research area in electrostatic catalysis of reactions. From a systematic study authors concluded that electrochemical non-idealities which have been considered as flaws are not flaws, and these non-idealities can be used for better understanding of semiconductor electrochemistry. Role of static positive charges of ferricenium ions on increasing reduction rate and decreasing oxidation rate is well explained (line 157-163) with energy level diagrams in Figure 2b. With this authors could explain the shift in anodic and cathodic peak positions. Conceptualization of such an experimental setup is exemplary.

This paper will be referred by many research groups whose expertise may not be in electrochemistry. However presentation of the paper is more of technical and difficult to follow by general readers. For example, shape of cyclic voltammetry in general is discussed in terms of size of diffusion layers and flux of ions towards the respective electrodes. General readers would get benefit from an explanation on parallel between the ferrocene modifications with size of diffusion layer and flux towards electrodes.

Introduction of this manuscript gave lot of attention to recent development in electrostatic catalysis of reactions, particularly the mechanistic developments. But the paper is not a study of reaction mechanism instead it deals more with the advance in electrochemical setup. So the readers should be well introduced to the relevant methodologies used in this paper which lack at this stage. The author should rewrite the introduction.

I feel authors have missed many important references regarding inverted peak found in earlier

literature e.g., *Electrochimica Acta* 52 (2007) 6378–6385 and *Journal of Electroanalytical Chemistry* 517 (2001) 15–19. To make the article more worthy to the community author should reconsider their ideas if they have any correlation or ambiguity with the previously reported literature. The Significance and Novelty of the article are high but the scholarly presentation and literature references should be improved.

Reviewer 1

Recommendation:

Reviewer 1: This paper report a very detailed study on the electrochemical behavior of molecular monolayer on Si/SiO₂ surface. Some findings from this work is interesting and convincing, including the role of static effects and the inversion of peaks in cyclic voltammograms. On the other hand, I have to conclude that it seems the paper may not be of interest to general audience, and I could not find sufficient significance for the observation reported in this paper. I have also some comments and suggestions.

Authors' reply: we thank the Reviewer for his/her very careful consideration of the manuscript. We agree that in the original submission a rather large focus was on observations of “reproducible flaws” during charge-transfer at semiconductors and its theoretical modelling. As a consequence, the immediate and more general implications of the work were definitely understated. Apart from performing extensive new experiments (*vide infra*) we have now re-written the Abstract, the Introduction and the Conclusion sections in order to present clearly and concisely where our findings will have immediate impact. In summary:

1) **Changes to Abstract and Introduction.** We have added remarks on the scope of a general electrode kinetic measurement and highlighted, at the very onset of the paper, that cyclic voltammetry is by far the prime and most widely-used form of electrochemical spectroscopy (i.e. simple, ubiquitous and informative).

We have re-written completely the introduction section accordingly to the remarks of Reviewer 1. The most relevant changes are discussed here. Firstly, we have stressed on the role that semiconductors have in society, technology and recent research:

“In 1876 Ferdinand Braun presented to a Natural Society meeting the first “deviations from the Ohm`s law” he had observed in crystals of galena, a natural form of lead sulphide. In the following century his discovery revolutionized our civilization. From galena to silicon, materials that can turn from conductors to insulators are at the basis of all our digitized technology.¹ Understanding the full spectrum of factors at play when charges are transferred across a semiconductor interface is crucial; it underpins the design of devices whose function span from converting light into electricity, to sensing their environment,² culminating in the very recent report of single-molecule rectifiers on Si(111).³”

For a general readership; we have now made clear that where there is the need to gain insights on charge transfer kinetics, either in energy conversion, catalysis or molecular electronics, then the scientist or the engineer require an analytical tool for the measurement. In this context, voltammetry still reigns supreme. Then, in order to translate experimental numbers into quantitative insights on the “speed” of charge transfer, any given electrochemical measurement needs to be coupled to a theoretical model.

Current theoretical models for the analysis of voltammetry fall short of capturing all of the factors at play, and most notably, they do not account for electrostatics. For semiconductors, and to a degree also for metals, the reproducibility of published kinetic data is extremely poor and we believe a big contributor to this problem is the naïve character of common models for kinetics. To stress on the importance of this we have added the following sentences to the Introduction:

“Chemically-modified electrodes are therefore a very important laboratory model system,⁴ however, a search of the literature will indicate that, in contrast to metallic electrodes, kinetic parameters for electron transfer at semiconductors are difficult to reproduce from laboratory to laboratory.^{5,6} Widely accepted approaches that are used to gain insights on electrode kinetics have clearly failed to reproduce the complex energetic landscape that determines the redox behavior observed. By highlighting the participation of “dynamic electrostatic” factors on charge transfer we are implicitly demonstrating that the electrostatic landscape of a silicon/molecular layer/electrolyte interface should either be accounted for or eliminated when the focus is on extracting kinetic data at semiconductor or photoconductor electrodes. This knowledge also opens up a semiconductor-based platform to aid the study of electrostatics on chemical reactivity,⁷⁻¹¹ and molecular electronics.^{3,12,13”}

We have now stated more explicitly that to either predict or manipulate charge transfer at the technologically most relevant material of today, i.e. silicon, will require scientists and engineers to account for and/or take precaution to limit interactions between “dynamic molecular charges” and the semiconductor substrate. From electro-catalysis to sensing and molecular electronics, this dynamic electrostatic landscape is always part of the system under investigation. It can be a “friend or a foe”, but either way it needs to be accounted for. This aspect was long neglected.

To substantiate this last point – i.e. *electrostatic interactions at redox molecular films are ubiquitous but most often overlooked* – and to reinforce further the significance and timeliness of our report, we note that in July 2017 Nature Nanotechnology published an article “breaking” the rectification record for molecular layers on metals (from ca 10^3 to ca 10^5) by means of harnessing the interactions between charged ferrocenes assembled as a molecular layer on a metal surface and the excess of surface charges on a contacting top electrode (X. Chen, et al., *Molecular diodes with rectification ratios exceeding 10^5 driven by electrostatic interactions*, **Nat. Nanotech.**, 12, 797–803 (2017), doi:10.1038/nnano.2017.110, Ref. 12 in the revised manuscript). The article is an elegant demonstration of the magnitude of electrostatic forces experienced by redox tethers at a gold “dry” electrode and it will no doubt stimulate similar high-impact lines of research in molecular electronics. However, the data and models presented in the manuscript allow, for the first time, scientists working with a more technologically relevant semiconducting material, such as silicon, and using more common “wet” systems (silicon electrode/molecular layer/electrolyte) to first detect, then quantify and eventually manipulate these effects. Si(111), one of the substrates used in our report, has just entered the area of single-molecule electronics, with STM single-molecule break-junctions in a solvent of low dielectric (mesitylene) made technically possible in 2017 (A.C. Aragonès, et al., *Single-molecule electrical contacts on silicon electrodes under ambient conditions*, **Nat. Commun.**, 8 (2017) 15056, Ref. 3 in the revised manuscript).

We anticipate that the models in our paper will now, for example, make it possible to assess very rapidly via cyclic voltammetry how factors such as solvent dielectric (i.e. electrostatic screening) or molecular length (i.e. kinetic factors and screening) play together defining the electrostatic forces experienced by an average molecule in a molecular layer. For instance, under the conditions where the attractions between molecules and semiconductor are gauged by voltammetry to be the largest, then the exact same sample could be analysed at the single-molecule level, with the rectification ratio expected to track the electrostatic measurement.

The new abstract is reproduced below using yellow highlight:

“Predicting or manipulating charge transfer at a semiconductor interface, from molecular electronics to energy conversion, relies on accessing knowledge that is generated from a kinetic analysis of the interfacial process. Cyclic voltammetry is by far the most frequently used technique for such studies, due to the combination of precise and simple control of potential (i.e. thermodynamics of the reaction) and sensitive measurement of current (i.e. kinetics of the process). Chemists, material scientists and engineers encountering non-ideal shapes and positions in voltammograms are often inclined to reject these as flaws. Here we show that at semiconductor electrodes these electrochemical non-idealities of surface confined redox probes (fwhm’s < 90.6 mV and anti-thermodynamic “inverted” peak potentials, $E_{\text{peak cathodic}} > E_{\text{peak anodic}}$) can be reproduced and are not flawed data. They are the manifestation of electrostatic interactions between charged redox molecules confined at a Si(111)/liquid interface and the semiconductor’s space-charge layer. A theoretical model is developed to decouple, under finite kinetics, the effect of electrostatics on the semiconductor side of the barrier (diode effects) from that on the activity of the surface-bound molecule (effects on intermolecular lateral interactions as described by Frumkin isotherm). This study highlights the interplay between diode and Frumkin parameters, and explains the impact of dynamic molecular charges on semiconductor electrochemistry. This has immediate general implications for a correct kinetic analysis of charge-transfer reactions at semiconductors as well as aiding the study of electrostatics on general chemical reactivity.”

”

2) **Conclusion:** Our original discussion on the impact our current findings will have on the rapidly emerging field of electrostatic catalysis (i.e. static charges guiding chemical non-redox reactivity) has been moved to the Conclusion section of the manuscript.

Reviewer: Concerning the selection of surface model system, what is the purpose to induce the long alkane chain? Will the length influence the conclusion?

Authors’ reply: We thank the Reviewer for highlighting this point. We have now conducted experiments using shorter alkyl chains (2- and 7-carbon spacers) to separate the redox unit from the semiconductor substrate. The findings of these experiments are reported below and have been a very valuable addition to the work.

Firstly, before the new data are presented and discussed, we would like to stress that the initial choice of the 9-carbon spacer (1,8-nonadiyne) was dictated by a careful consideration on the trade-off between the kinetics of charge transfer across the layer (tunnelling distance) versus the stability and order of the

molecular layer. We discovered that modifying the surface with 1,8-nonadiyne leads to highly-ordered molecular layers (S. Ciampi, J.B. Harper, J.J. Gooding, *Wet chemical routes to the assembly of organic monolayers on silicon surfaces via the formation of Si–C bonds: surface preparation, passivation and functionalization*, **Chem. Soc. Rev.**, 39 (2010) 2158-2183, Ref. 24 in the revised manuscript) and upon derivatization, the redox response of the electrode is close-to-ideal redox. This “wet” chemistry approach is far superior than had ever previously been reported (S. Ciampi, P.K. Eggers, G. Le Saux, M. James, J.B. Harper, J.J. Gooding, *Silicon (100) Electrodes Resistant to Oxidation in Aqueous Solutions: An Unexpected Benefit of Surface Acetylene Moieties*, **Langmuir**, 25 (2009) 2530-2539, Ref. 25 in the revised manuscript).

An alternative approach would be to pre-synthesise an alkynyl-ferrocene and then to assemble this molecule over the substrate in a single-step (see B. Fabre, *Ferrocene-Terminated Monolayers Covalently Bound to Hydrogen-Terminated Silicon Surfaces. Toward the Development of Charge Storage and Communication Devices*, **Acc. Chem. Res.**, 43 (2010) 1509-1518 or J. J. Gooding, S. Ciampi, *The Molecular Level Modification of Surfaces: From Self-Assembled Monolayers to Complex Molecular Assemblies*, **Chem. Soc. Rev.**, 40 (2011) 2704-2718). However, the monolayers formed with the pre-synthesised method are generally of significantly inferior quality. For example, we have now tested the direct attachment of ethynylferrocene onto the hydrogenated silicon surface (as an example of very short, 2-carbon spacer). We used two different synthetic methods for the hydrosilylation of ethynylferrocene: either assisting the reaction with UV light or by supplying heat. In both cases the electrode surface is rapidly oxidized during the potential scanning (e.g. Figure A), which again points to the importance of using a longer alkyl chain as a protective layer which can then be chemically functionalized.

Figure A. *UV-assisted hydrosilylation of ethynylferrocene on Si(111)-H.* Direct grafting of ethynylferrocene (i.e. an example of a 2-carbon spacer) onto the hydrogenated Si(111) substrate leads to poorly-behaved electrodes. Rapid oxidation of the silicon surface is evident and the current intensity drops rapidly under a potential ramp. This pre-synthesized (one-step) approach would bring the ferrocene unit in very close proximity of the semiconductor but unfortunately it falls short of being a satisfactory “model substrate” to probe the “dynamic electrostatic” models of the paper. Cyclic voltammogram (100 mV/s) in 1.0 M HClO₄ under supra band gap illumination.

Besides having an impact on the chemical stability of the silicon surface, the alkane chain length will impact the kinetics of the electron transfer, as this decreases with increasing chain length (Fabre B., *Functionalization of Oxide-Free Silicon Surfaces with Redox-Active Assemblies*, **Chem. Rev.**, 116 (2016) 4808-4849, Ref. 5 in the revised manuscript), which brings to the interesting question raised by the Reviewer. We would expect a shorter aliphatic chain to decrease the dielectric screening, and therefore enhance the sensing by the linked ferrocene of the substrate electrostatics.

As pointed out above, there are some limitations on the extent to which we can shorten the alkyl spacer. We therefore tried forming monolayers on the atomically flat Si(111) surface using 1,6-heptadiyne (7-carbon spacer). The hydrosilylation of 1,6-heptadiyne on Si(111)-H and subsequent “click” grafting of ferrocene, as done for the 1,8-nonadiyne layers of the original manuscript, led to a decrease of *ca.* 2 Å of the experimental (XRR-determined, *vide infra*) ferrocene-silicon distance (see Figures **B** and **C**). The monolayers obtained from both α,ω -diynes molecules led to similar coverage (electron densities as determined by XRR, see Tables **A** and **B** below). Figure **B** compares voltammograms obtained for ferrocene-modified surfaces prepared from either 1,8-nonadiyne or 1,6-heptadiyne. It is apparent that the electrostatic sensing of the space charge is enhanced by using a shorter alkyl chain as can be observed from the narrowing of the peaks (78 mV, fwhm) and the onset of inverted peak potentials ($E_{p,a}-E_{p,c}=-30$ mV). This is a remarkably interesting result and it validates our hypothesis: it shows that the electrostatic interactions scale with the inverse of the distance, highlighting the importance of seemingly minor experimental changes (*i.e.* a 2 Å decrease in distance). However, for these new experiments, as the chain gets shorter, the molecular layer becomes unfortunately very short-lived. We cannot access further quantitative insights on the 1,6-heptadiyne system since a systematic change in voltage sweep rates is accompanied by rapid loss of surface coverage. New synthetic methodologies will probably need to be explored before something more definitive on the practical limits of these ~ 5 Å thick carbonaceous films (C-7) can be drawn.

A note on these experiments was added in page 4 of the main text (yellow highlight, footnote number 30).

Figure B. Sensing the electrostatic interaction between the space charge layer and the ferrocene by cyclic voltammetry. Electrostatic interactions manifests as “peak potential inversion” as well as peak narrowing. Both are enhanced by shortening the ferrocene-silicon distance going from a 1,8-nonydiyne to a 1,6-heptadiyne base layers. XRR-determined values of monolayer thickness after covalent attachment of ferrocene units are depicted by labels in figure. The substrate is Si(111) and representative voltammograms are shown in figure (100 mV/s in 1.0 M HClO₄ electrolyte under illumination). The electrochemically-determined ferrocene coverage is $\Gamma=5.2 \cdot 10^{-11}$ mol·cm⁻² for the nonadiyne-Fc (left) and $\Gamma=1.8 \cdot 10^{-10}$ mol·cm⁻² for the heptadiyne-Fc (right) system.

Table A. Results from fits to X-ray reflectometry data for the Si(111) samples modified with 1,6-heptadiyne-ferrocene.

Layer	Thickness / Å	SLD / $\times 10^{-6}$ Å ⁻²	Roughness / Å
Air	-	0 *	-
Organic monolayer	13.3 ± 0.1	12.45 ± 0.06	0.2 ± 0.1
Silicon substrate	-	20.1 *	0.3 ± 0.2

Table B. Results from fits to X-ray reflectometry data for the Si(111) samples modified with 1,8-nonadiyne-ferrocene.

Layer	Thickness / Å	SLD / $\times 10^{-6} \text{ \AA}^{-2}$	Roughness / Å
Air	-	0.0 *	-
Organic monolayer	15.0 ± 0.2	11.9 ± 0.1	1.2 ± 0.1
Silicon substrate	-	20.1 *	3.1 ± 0.1

NOTE: SLD (scattering length density) for X-rays is obtained by multiplying the electron density ($e^{-}/\text{\AA}^3$) of the material by the factor $2.82 \times 10^{-5} \text{ \AA}$.

* The SLD for air and silicon are known and were fixed parameters.

Figure C. X-ray reflectometry. Experimental monolayer thickness, electron density and roughness. Comparison of XRR interference fringes on heptadiyne-Fc (green) and nonadiyne-Fc (blue) Si(111) samples. The shift in the XRR fringe to lower Q in the blue curve shows that the nonadiyne-derived film is a thicker monolayer. XRR data were refined against theoretical models and structural parameters are summarized in the Tables above (extensive experimental details are in the revised SI section).

Reviewer: The z scale of Figure 3a-c the surface is missing. Concerning the coverage calculation, as suggested by the AFM images, the surface seems quite rough. I was wondering whether the surface roughness calculation from average root mean square is valid for this case. Some control experiment may be required.

Authors' reply: We apologize for the absence of a z-axis scale from the original figure. The scale has been added to Figure 3 of the revised manuscript. For clarity and comparison purposes, all the AFM images were normalized to 1.4 nm. In addition to a z-axis scale we have also explicitly listed in caption the most relevant AFM-derived topographical data. Furthermore, we have validated AFM data against X-ray reflectometry measurements of substrate-monolayer and monolayer-air roughness.

Figure 3: Anodic monolayer “stripping”. Tapping mode AFM images ($2 \times 2 \mu\text{m}$) and XPS Si 2p narrow scans for **S-2** samples on Si(111). Cyclic voltammograms are shown as insets to the AFM images. Data for “as-prepared” samples are in (a) and (d), and data after the short potential step (0.3 V, 1 min) that leads reproducibly to narrow voltammetric waves (ca. 60 mV fwhm) are in (b) and (e). Due to kinetic factors the narrow waves are lost upon more extensive oxidation of the electrodes ((c) and (f)). High-energy emissions from silicon oxides ($\text{Si}^+-\text{Si}^{4+}$) are evident in (e) and (f), and the evolution of the $\text{SiO}_x:\text{Si}$ 2p peak area ratios appear as labels in the XPS panels. (a) Small protrusions are observed (222 protrusions/ μm^2 , which are on average 0.4 nm in height and 5 nm in radius) over the staircase structure of the $\langle 111 \rangle$ surface (R_q of 0.08 ± 0.02 nm ($R_{\text{tm}}=0.15$ nm) on the terraces and $R_q=0.20 \pm 0.15$ nm ($R_{\text{tm}}=0.63$ nm) over the whole area). The number of protrusions increases upon the anodic treatment (327 protrusions/ μm^2 in (b) and 945 protrusions/ μm^2 in (c)) and tracks XPS SiO_x emissions. For clarity and comparison purposes all the AFM images are normalized to 1.4 nm.

With regards to the Reviewer's remark "*the surface seems quite rough*", firstly we hope the AFM z -axis and the value of R_q (root mean square, $R_q = 0.08 \pm 0.02$ nm on the terraces and $R_q = 0.20 \pm 0.15$ nm over the whole area) help showing that this is not the case. In addition to values of R_q we have now specified the value of peak-to-valley roughness (R_{im} in short hand, i.e. average distance between highest peak and lowest valley) on the individual $\langle 111 \rangle$ terraces as well as for the entire area scanned. The peak-to-valley roughness is 0.15 ± 0.02 nm on individual terraces or 0.63 ± 0.06 nm over the entire area. These values are now explicitly stated in Figure 3. The substrates appear therefore to be near to atomically flat, which is an indication of the high quality of the surface samples (Wallart X, Henry de Villeneuve C, Allongue P. *Truly Quantitative XPS Characterization of Organic Monolayers on Silicon: Study of Alkyl and Alkoxy Monolayers on H-Si(111)*. **J. Am. Chem. Soc.** 2005, 127(21), 7871-8, Ref. 33 in the revised manuscript, Aswal DK, Lenfant S, Guerin D, Yakhmi JV, Vuillaume D., *Self assembled monolayers on silicon for molecular electronics*, **Anal. Chim. Acta**, 568 (2006) 84-108, Ref. 34 in the revised manuscript and Faucheux A, Gouget-Laemmel AC, Henry de Villeneuve C, Boukherroub R, Ozanam F, Allongue P, et al., *Well-Defined Carboxyl-Terminated Alkyl Monolayers Grafted onto H-Si(111): Packing Density from a Combined AFM and Quantitative IR Study*, **Langmuir**, 22 (2006) 153-162, Ref. 35 in the revised manuscript).

As mentioned briefly above, in the revision we have performed additional measurements of surface roughness by using an independent technique, X-Ray Reflectometry (XRR). The roughness obtained by XRR (0.3 nm for the monolayer/silicon interface of **S-2** samples on Si(111)) is entirely consistent with the AFM data. The following concise statement was added to the manuscript (page 6):

"The surface roughness is extremely low which is an indication of the high quality of the surface samples (*ca.* 0.2 nm peak-to-valley roughness measured by AFM on individual terraces, details in Figure 3, or a *ca.* 0.3 nm overall roughness determined independently by X-ray reflectometry, Supplementary Information section S7).³³⁻³⁵"

Details on the technique and on how structural parameters for these monolayer systems are extracted from the refinement of XRR data have been added to the manuscript and Supplementary Information. These are summarized in yellow highlight below:

Added in the main text, under **Methods/Chemicals and Materials/Surface/Characterization/Specular X-ray Reflectometry (XRR)**.

Specular X-ray Reflectometry (XRR). XRR measurements were measured under ambient conditions on a Panalytical Ltd X'Pert Pro X-ray Reflectometer using Cu $K\alpha$ X-ray radiation ($\lambda = 1.54056$ Å). The X-ray beam was focused using a Göbel mirror and collimated with 0.1 mm pre-sample slit and a post-sample parallel plate collimator. Reflectivity data were collected over the angular range $0.05^\circ < \theta < 5.00^\circ$, with a step size of 0.010° and counting times of 7 seconds per step. The data was reduced by normalising the data so that the critical edge was unity and was presented as reflectivity (= reflected intensity / incident intensity) vs. momentum transfer (Q) which is equal to $Q = (4\pi \sin \theta) / \lambda$, where θ is the angle of incidence and λ is the wavelength. Structural parameters for the monolayer were refined in MOTOFIT reflectometry analysis software.⁶¹

Added in the Supplementary Information, under **S7. Surface characterization –XRR methods and data**

Structural parameters for the monolayer were refined in MOTOFIT reflectometry analysis software.³¹ The monolayer is conceptualised as a layered system with each layer defined by its thickness, X-ray scattering length density (SLD), and interfacial roughness. MOTOFIT utilises the Abeles matrix formalisation to calculate the specular reflectivity from a stratified layer system. In this method the system is split into a series of layers and the incident radiation is refracted by each layer. The value of the wave vector (k) in layer n is given by:

$$k_n = \sqrt{k_0^2 - 4\pi(\rho_n - \rho_0)}$$

Where $k_0 = Q/2$ and ρ is the SLD of the layer. The Fresnel reflection coefficient ($r_{n,n+1}$) between layers n and $n+1$ is described by:

$$r_{n,n+1} = \frac{k_n - k_{n+1}}{k_n + k_{n+1}} \exp(-2k_n k_{n+1} \sigma_{n,n+1}^2)$$

The term after the exponential function in the above equation is a Gaussian error function to account for the roughness between each interface.^{32,33} The Fresnel reflection coefficients along with phase factors are used to calculate a characteristic matrix for each layer, the product of which is used to calculate the reflectivity. A least-square fitting routine is used to minimise χ^2 values between observed and calculated reflectivity using a genetic algorithm. The fitting of the reflectometry profile yields information on the SLD profile normal to the surface. The SLD can be considered as an X-ray refractive index and is a function of the chemical composition of the material according to:

$$SLD = \frac{\sum_{i=1}^n Z_i r_e}{V_m}$$

Where Z_i is the atomic number of the i th atom, r_e is the Bohr electron radius (2.818×10^{-15} m), and V_m is the molecular volume determined to be 375 \AA^3 using the web tool Molinspiration (<http://www.molinspiration.com/>). From this the theoretical SLD of a complete monolayer is determined to be $14.2 \times 10^{-6} \text{ \AA}^{-2}$. By comparing the theoretical SLD to the fitted SLD of the monolayer the volume fraction (ϕ) of the monolayer can be determined as:

$$\phi = \frac{SLD_{fitted}}{SLD_{theoretical}}$$

Using the volume fraction and other parameters the surface excess (Γ) in mol cm⁻² can be calculated as follows:

$$\Gamma = \frac{\varphi\tau 10^{16}}{V_m N_A}$$

Where τ is the monolayer thickness and N_A is Avogadro's constant.

Measuring the X-ray reflectometry of the monolayer on silicon under ambient conditions showed a single broad fringe approximately $0.2 < Q < 0.6 \text{ \AA}^{-1}$ indicating that a well formed monolayer had been deposited onto the silicon substrate (Supplementary Figure 23A). The best fit to the X-ray reflectometry data was using a model where the monolayer system is defined as a single layer (red line in Supplementary Figure 23A). The total thickness of the monolayer was determined to be 14.9 \AA (Supplementary Table 1) which is consistent with previous monolayer systems that utilise the same click chemistry method for fabrication. The fitted SLD was found to be $11.9 \times 10^{-6} \text{ \AA}^{-2}$ and this corresponds to a surface excess of $5.56 \times 10^{-10} \text{ mol cm}^{-2}$ which is consistent with the determination of surface coverage from the electrochemical data. The system consists of two interfaces at the air/monolayer interface and monolayer/silicon interface each with a roughness of 1.2 \AA and 3.1 \AA respectively (Supplementary Table 1 and Supplementary Figure 23B). The X-ray reflectometer was set up to illuminate a relatively large area of $10 \times 10 \text{ mm}^2$ and therefore provides a global view of roughness compared to AFM which sampled over a much smaller area of $2 \times 2 \text{ \mu m}^2$ providing a localised view of roughness determined to be 2.0 \AA . The roughness values between the two independent techniques are consistent showing that the sample area chosen for AFM is typical across the larger area.

Supplementary Table 1. Results from fits to X-ray reflectometry data for the Si(111) samples modified with 1,8-nonadiyne-ferrocene.

Layer	Thickness / \AA	SLD / $\times 10^{-6} \text{ \AA}^{-2}$	Roughness / \AA
Air	-	0.0 *	-
Organic monolayer	14.9 ± 0.2	11.9 ± 0.1	1.2 ± 0.1
Silicon substrate	-	20.1 *	3.1 ± 0.1

NOTE: SLD (scattering length density) for X-rays is obtained by multiplying the electron density ($e/\text{\AA}^3$) of the material by the factor $2.82 \times 10^{-5} \text{ \AA}$.

* The SLD for air and silicon are known and were fixed parameters.

Supplementary Figure 23. X-ray reflectometry. Experimental monolayer thickness, electron density and roughness. A) X-ray reflectivity profile of the S-2 monolayer system on Si(111). The points with error bars are the collected data and the red line is the fit to the data. B) Real-space SLD profile of the monolayer systems with distance 0 \AA set as the interface between air and the monolayer. The vertical dashed lines show the boundary between the air/monolayer and monolayer/silicon interfaces.

Reviewer: In figure 6, it seems almost not data point presented in the QR regime, why? Some further discussion will help.

Authors' reply: This is a valuable remark. The purpose of Figure 6 is to show how the electrostatic effect that brings the anodic wave to cathodic potentials, **although always present**, is often masked by the kinetics of the electron transfer. When the effective charge transfer constant rate is high enough the electrostatics effect is apparent (R region). The boundaries of the transition region (QR region of the original submission) were arbitrarily chosen, and they were only intended to show where the kinetics starts to mask the electrostatic effect on the “peak potential inversion”. To make sure the data in the figure effectively conveys this message (“inversion” always present but masked by slow kinetics) we have now removed the QR region and only drawn an arbitrary boundary between a reversible region (fast kinetics, R, with a linear dependence between peak currents and the scan rate and practically identical slopes for cathodic and anodic scans) and a non-reversible region (slow kinetics, NR). The new Figure 6 of the main text is reproduced below for clarity:

Figure 6: Kinetic regions for the voltammetry as a function of the scan rate. Evolution of the experimental peak potentials (E_{peak} , (a)) and currents (I_{peak} , (b)) as a function of voltage sweep rate (v) in the cyclic voltammetry of S-2 samples on a-Si. Black and white symbols correspond to the anodic and cathodic data, respectively. An arbitrary boundary between reversible (fast kinetics, R), and non-reversible (finite kinetics, NR) regions is indicated in figure. Lines are a guide to the eye only.

Furthermore, Figure 6 in the main text is one example of “inverted” system. The phenomena is always observed in amorphous silicon, as now stated at page 11, lines 225-230, of the revised manuscript (see also the additional independent experiments reproduced below). The range of voltage sweep rates is very wide (from 0.001 V/s to 1.5 V/s) and the presence of relatively few data points in Figure 6 between 0.1 V/s and 0.25 V/s bears no practical significance on overall data analysis. We have added these sets of data to the Supplementary Information, Supplementary Figure 16 and made explicit reference in the manuscript on the approximate location of the R/NR boundary, footnote reference 40.

Supplementary Figure 16. Kinetic regions for the voltammetry as a function of the scan rate. Evolution of the experimental peak potentials (E_{peak} , (a)) and currents (I_{peak} , (b)) as a function of voltage sweep rate (v) in the cyclic voltammetry of **S-2** samples on a-Si. Black and white symbols correspond to the anodic and cathodic data, respectively. Reversible (fast kinetics, R) and non-reversible (finite slow kinetics, NR) zones are arbitrarily indicated in figure. Lines are a guide to the eye only.

Reviewer: I cannot quite follow the schematic of Figure 3b and 5a, especially the E vs reference arrow, what changed?

Authors' reply: we believe the Reviewer is referring to the band-bending diagrams in Figures 2b and 5a. This type of schematic is often used in physics and chemical sciences and we have followed the convention of drawing the available energy-levels such that the energy of the system is lowered when electrons fall down toward the positive potential (a similar statement is found at page 7, lines 153 and 154 of the manuscript). To ensure our interpretation of the electrostatic role on redox kinetics was correct, we reasoned that what holds for photoanodes (n-type, Figure 2) should have its counterpart in photocathodes (p-type, Figure 5). The symmetry change of band-bending versus *E* reference arrow in Figures 2 and 5 (i.e. tilting up or down) gives a visual and schematic depiction of the photoanodes and photocathode systems. We have made more explicit reference to this in Figure 2:

Figure 2, “(b) Distorsion of the semiconductor-side of the barrier **for a photoanode** due to the presence of an electrochemically-induced dipole layer of surface charges”

The model therefore holds for both n- and p- type semiconductors, which is the simplest and most direct test of the consequences of our electrostatic models on charge transfer at interfaces (page 10, lines 208-211, of the revised manuscript).

Reviewer: To conclude, I think this paper is interesting but may not provide enough significance for publication in nature communications. I have the feeling that this paper could be more suitable for a more specific journal.

Authors' reply: As stated in our reply to the reviewer's first question, we have included in the revised manuscript experiments and relevant text to highlight the significance of our results to engineers and scientists. The intended target audience of this work is indeed very large. There are very few branches of chemistry, physics and engineering that do not deal with interfaces and an extremely large proportion of these deal with (deliberately or not) electrified interfaces. As discussed in the opening of this letter, we acknowledge that our initial submission was understating these general aspects. We have now revised the introduction to be accurate, brief, and clear and at the same time engage the largest audience of scientists and engineers, from energy, to sensing and molecular electronics.

Our initial submission had a large focus on the chemical implications of electrostatics. Some recent developments on this are now discussed in the concluding remarks while some others are summarized here below. The relevance of this branch of chemistry was greatly appreciated by Reviewer 2, but we agree with 1 that more readers will find the paper immediately relevant to them when semiconductors / failure & limits of currently-available theoretical models / electrostatic effects are laid out in this order.

We need to stress that what was till about 2 year ago considered a theoreticians “day dreaming”, i.e. a link between electrostatics and chemical reaction rates and selectivity, has just recently been experimentally verified. The chemical implications of static charges are gaining unprecedented attention, with the impact of the topic being reflected by several high-profile articles published over the last 12 months. Examples include C. Geng, J. Li, T. Weiske, M. Schlangen, S. Shaik, H. Schwarz, *Electrostatic and Charge-Induced Methane Activation by a Concerted Double C–H Bond Insertion*, **J. Am. Chem. Soc.**, 139 (2017) 1684-1689, Ref. 11 in the revised manuscript; S. Shaik, D. Mandal, R. Ramanan, *Oriented electric fields as future smart reagents in chemistry*, **Nat. Chem.**, 8 (2016) 1091-1098, Ref. 7 in the revised manuscript; A.C. Aragonès, N.L. Haworth, N. Darwish, S. Ciampi, N.J. Bloomfield, G.G. Wallace, I. Diez-Perez, M.L. Coote, *Electrostatic catalysis of a Diels–Alder reaction*, **Nature**, 531 (2016) 88-91, Ref. 10 in the revised manuscript.

As alluded by Reviewer 2, now the challenge is to develop experimental platforms to scale from a few hundred thousand chemical events under field (Ref. 10 in the revised manuscript) to billions of billions of events. We believe the knowledge this manuscript contributes to measure the electrostatic “landscape” of a semiconductor/monolayer/electrolyte will be important in this context. Metal/organic/metal and metal/organic/electrolyte junctions have been extensively studied (see for example: •Yan H, Huang Q, Cui J, Veinot JGC, Kern MM, Marks TJ, *High-Brightness Blue Light-Emitting Polymer Diodes via Anode Modification Using a Self-Assembled Monolayer*, **Adv. Mater.**, 15 (2003) 835-838 • de Boer B, Hadipour A, Mandoc MM, van Woudenberg T, Blom PWM, *Tuning of Metal Work Functions with Self-Assembled Monolayers*, **Adv. Mater.**, 17 (2005) 621-625 • Campbell IH, Kress JD, Martin RL, Smith DL, Barashkov NN, Ferraris JP, *Controlling charge injection in organic electronic devices using self-assembled monolayers*, **Appl. Phys. Lett.**, 71 (1997) 3528-3530), however the study of semiconductor/organic/electrolyte junctions is an exciting field. Here we have developed a model to isolate the electrostatic contribution to electron transfer in current-potential curves. The knowledge generated from this study will be of interest not only in fields such as semiconductor electronics, sensing and charge transfer but also in the emerging field of “electrostatic catalysis”.

Reviewer 2

Recommendation:

Reviewer 2: *A report on “Reproducible flaws unveil electrostatic aspects of semiconductor electrochemistry” Nature Communication.*

This paper reports some important science that can be a proper guidance and will be a reference for an emerging research area in electrostatic catalysis of reactions. From a systematic study authors concluded that electrochemical non-idealities which have been considered as flaws are not flaws, and these non-idealities can be used for better understanding of semiconductor electrochemistry. Role of static positive charges of ferricenium ions on increasing reduction rate and decreasing oxidation rate is well explained (line 157-163) with energy level diagrams in Figure 2b. With this authors could explain the shift in anodic and cathodic peak positions. Conceptualization of such an experimental setup is exemplary.

This paper will be referred by many research groups whose expertise may not be in electrochemistry. However presentation of the paper is more of technical and difficult to follow by general readers. For example, shape of cyclic voltammetry in general is discussed in terms of size of diffusion layers and flux of ions towards the respective electrodes. General readers would get benefit from an explanation on parallel between the ferrocene modifications with size of diffusion layer and flux towards electrodes.

Introduction of this manuscript gave lot of attention to recent development in electrostatic catalysis of reactions, particularly the mechanistic developments. But the paper is not a study of reaction mechanism instead it deals more with the advance in electrochemical setup. So the readers should be well introduced to the relevant methodologies used in this paper which lack at this stage. The author should rewrite the introduction.

*I feel authors have missed many important references regarding inverted peak found in earlier literature e.g., *Electrochimica Acta* 52 (2007) 6378–6385 and *Journal of Electroanalytical Chemistry* 517 (2001) 15–19. To make the article more worthy to the community author should reconsider their ideas if they have any correlation or ambiguity with the previously reported literature.*

The Significance and Novelty of the article are high but the scholarly presentation and literature references should be improved.

Authors’ reply: we thank Reviewer 2 for the appreciation of the work and very fruitful comments. The remarks of 2 on the structure of the Introduction are very valuable ones. The Abstract, Introduction and Conclusion sections have been now largely re-written. The newly-written Introduction should engage a larger audience – from the surface scientist working in energy conversion to the engineer, chemist or physical chemist whose interest or laboratory model pivots around an electrified interface. The main changes are explained in detail in the response to 1 (*vide supra*). In brief, emphasis is now been placed on the general use and ubiquity of voltammetry. This remains the prime analytical tool to probe into the mechanism through which ionic conduction turns in electronic conduction – i.e. the essence of electrode kinetics –. We have then made explicit references to the scope of an electrode kinetic measurement, we have listed the assumptions and limits of current theoretical models used to describe the electrified interface and we have highlighted the lack of (correct) knowledge on electron transfer and electrostatic effects on semiconductor electrodes. These effects should either be accounted for or precautions should be taken to limit them.

In summary, the new Introduction is focused on a broader discussion of electrostatic, electrode kinetics, and semiconductors. In the originally-submitted manuscript we opted to present at first the chemical implications of electrostatics. We thank Reviewer 2 for stressing that the current work is expected to provide a new platform with implications well beyond electrochemistry (“*This paper will be referred by many research groups whose expertise may not be in electrochemistry*” and “*will be a reference for an emerging research area in electrostatic catalysis of reactions*”).

Reviewer 2: I feel authors have missed many important references regarding inverted peak found in earlier literature e.g., Electrochimica Acta 52 (2007) 6378–6385 and Journal of Electroanalytical Chemistry 517 (2001) 15–19. To make the article more worthy to the community author should reconsider their ideas if they have any correlation or ambiguity with the previously reported literature.

Authors' reply: this point warrants a clarification. It is true that term 'peak inversion' could eventually mislead the general reader. However, the 'inverted peaks' of the literature Reviewer 2 is referring to relate to the inversion of currents in cyclic voltammograms, i.e. a cathodic peak that appears on the anodic scan (or vice versa). This phenomena arise from the presence of complex redox processes which combine adsorption and mass transport (i.e., diffusion) steps. The joint influence of "adsorptive" and "diffusive" controls is the key detail here. In our "diffusionless" system we referred to the term 'peak inversion' not as an inversion on peak currents but on the inversion of peak potentials, i.e. the anodic wave appears at potentials more cathodic than the cathodic wave. This is a totally different phenomena; the shift in the peak potential reflects the energetics of the system being altered due to the presence of electrostatic charges. Nonetheless, Reviewer 2 is entirely correct and the term 'peak inversion' will certainly bring confusion to some readers. We replaced it by the term 'peak potential inversion'. We also added a footnote (Ref. 37 in the revised manuscript) on page 8 clarifying on this point as well as to make reference to the literature cited by Reviewer 2.

REVIEWERS' COMMENTS:

Reviewer #1 (Remarks to the Author):

I am pleased that the author answered the questions and revised the manuscript carefully. The significance becomes clear after rewriting the abstract and introduction. Despite the study seems too specific to general audience, the quality improves significantly to meet the standard of the Nature communications. Accordingly, I agree this paper to be published on Nature communications.

Reviewer #2 (Remarks to the Author):

The authors have revised their manuscript based on my review as well as the one by another reviewer.

I read the paper twice and find it much improved. Now it fits the general reader. I also read the rebuttal to the second referee. I found the response to be remarkably good and professional. I therefore recommend acceptance of the papers subject to two minor revisions:

- a) The abstract still contains technical jargon ["The interplay between diode and Frumkin parameters"]. I am confident that most readers have no clue what are Frumkin's parameters. There must be a way to avoid this technical [albeit important] term in the abstract.
- b) Page 15: Here a sentence with inaccurate citation [": with models developed and refined by Shaik and others starting from 1981, [refs:9,45-49"]]. It should be: : with models developed and refined by Shaik[ref. 8] and others starting from 1981[refs:9,45-49].

Dr Simone Ciampi

Department of Chemistry
Curtin University
Bentley WA 6102 Australia

Telephone: +61 8 9266 9009
Facsimile: +61 8 9266 2300
Email: simone.ciampi@curtin.edu.au

October 25th, 2017

Dear Editor,

Many thanks for the exciting news about our **manuscript NCOMMS-17-03666** entitled ‘*Reproducible flaws unveil electrostatic aspects of semiconductor electrochemistry*’ by Yan B. Vogel, Long Zhang, Nadim Darwish, Vinicius R. Gonçales, Anton Le Brun, J. Justin Gooding, Angela Molina, Gordon G. Wallace, Michelle L. Coote, Joaquin Gonzalez, & Simone Ciampi.

We have now addressed the last remarks of **Reviewer 2** and revised accordingly the concluding section of the manuscript.

Reviewer 1

***Reviewer 1:** I am pleased that the author answered the questions and revised the manuscript carefully. The significance becomes clear after rewriting the abstract and introduction. Despite the study seems too specific to general audience, the quality improves significantly to meet the standard of the Nature communications. Accordingly, I agree this paper to be published on Nature communications.*

Authors’ reply: we thank **Reviewer 1** for his suggestions, valuable time and appreciation of the work.

Reviewer 2

***Reviewer 2:** The authors have revised their manuscript based on my review as well as the one by another reviewer. I read the paper twice and find it much improved. Now it fits the general reader. I also read the rebuttal to the second referee. I found the response to be remarkably good and professional. I therefore recommend acceptance of the papers subject to two minor revisions:
a) The abstract still contains technical jargon ["The interplay between diode and Frumkin parameters"]. I am confident that most readers have no clue what are Frumkin's parameters. There must be a way to avoid this technical [albeit important] term in the abstract.*

Authors’ reply: The Abstract was edited to remove technical jargon from it. Any specific reference to Frumkin’s parameters has been avoided.

***Reviewer 2:** b) Page 15: Here a sentence with inaccurate citation [": with models developed and refined by Shaik and others starting from 1981, [refs:9,45-49"]]. It should be: with models developed and refined by Shaik[ref. 8] and others starting from 1981[refs:9,45-49].*

Authors' reply: The citation has now been fixed in the revised manuscript by making a clear distinction between Shaik`s work and the work of others. The statement now reads as:

“with models developed and refined by Shaik⁴²⁻⁴⁵ and others^{9,46} starting from 1981”

We hope that the manuscript is now suitable for publication. Thank you for your time and looking forward to hearing from your editorial office.

Yours sincerely,

Simone Ciampi